# Effects of an African Circle Dance Programme on Internally Displaced Persons with Depressive Symptoms: A Quasi-Experimental Study

**DOI:** 10.3390/ijerph18020843

**Published:** 2021-01-19

**Authors:** Dauda Salihu, Eliza M. L. Wong, Rick Y. C. Kwan

**Affiliations:** Centre for Gerontological Nursing, School of Nursing, The Hong Kong Polytechnic University, Kowloon, Hong Kong 999077, China; dauda.salihu@connect.polyu.hk (D.S.); eliza.wong@polyu.edu.hk (E.M.L.W.)

**Keywords:** internally displaced persons, African circle dance, stress, anxiety, depressive symptoms

## Abstract

*Background*: Internally Displaced Persons (IDPs) are people who have been forced to flee their homes due to disasters. Depressive symptoms, at over 31–67%, are prevalent in IDPs in Africa. Despite the evidence for the benefits of the promotion of dance interventions on psychological health, supporting information is needed to outline the benefits of an African Circle Dance (ACD) intervention for IDPs in Africa. *Methods*: A quasi-experimental design (pre-/post-test) was employed. Two IDP camps were randomized into the intervention group (psychoeducation and ACD intervention) and the control group (psychoeducation). Adults aged ≥18 years, living in an IDP camp, able to perform brisk walking, and who scored ≥10 on a depressive symptoms subscale were recruited. The intervention group received an 8-week ACD dance intervention and two 1-h psychoeducation sessions on stress management; the controls only received the psychoeducation sessions. Outcomes were depressive symptoms, stress, and anxiety. Data were collected at baseline (T0), immediately after the intervention at week 8 (T1), and at week 12 (T2) at the post-intervention and follow-up session. A generalized estimating equation was used to test the effects of the ACD intervention, with a 0.05 significance level. *Results*: 198 IDPs completed the study (n_control_ = 98; n_intervention_ = 100). The intervention group reported significantly greater improvements in depressive symptoms (*v* = 0.33, *p* < 0.001) and stress (*v* = 0.15, 0.008) than did the control group. *Conclusions*: ACD could be a valuable complementary intervention in health promotion but more research is needed.

## 1. Introduction

Internally Displaced Persons (IDPs) are defined as people who are forced by disasters to flee their places of residence and go elsewhere within the borders of their country [1]. The internal displacement of people through violence is a globally prevalent issue that affects approximately 45.7 million people, equivalent to around 0.6% of the world’s population [2]. They are more frequently found in several countries, including Syria, Democratic Republic of Congo, Ethiopia, Burkina Faso, and Afghanistan [2]. A majority of them are African [3]. Displaced people face numerous challenges, including psychological trauma; however, mental health problems (e.g., stress, anxiety, and depressive symptoms) are commonly neglected [4].

Traumatic experiences are inevitable throughout human life [5]. Traumatic experiences in some populations are often the result of disasters such as conflicts, tsunamis, and earthquake [6]. Recently, internal displacement as a result of conflicts is increasingly prevalent [7]. In general, evidence has shown that victims of traumatic experiences were prone to psychoemotional problems such as anxiety and depressive symptoms [8].

Depressive symptoms are the manifestation of sadness, disruptive patterns of sleep, poor appetite, depressed mood, loss of pleasure or interest, and low self-worth or feelings of guilt, resulting in impaired functioning or psychological distress [9]. Stress is strongly associated with the development of depressive symptoms [10]. When left unmanaged, stress could cause anxiety [11]. Depressive symptoms are found in 5–80% of globally dispersed IDPs [12], with 31–67% in Africa, and 80.3% in Asia [3,13]. Such symptoms are associated with functional impairment, low quality of life, short life expectancies, substance abuse, reduced work productivity, and suicidal deaths [14].

To date, conventional treatments for depressive symptoms had depended on pharmacological approaches, despite the known association between the use of drugs and adverse effects, including erectile dysfunction, anxiety, and insomnia [15]. Non-pharmacological approaches are arguably preferable because of their limited adverse effects [16]. Non-pharmacological strategies involving psychological health interventions such as psychotherapy (e.g., cognitive behavioural therapy), physical health interventions (e.g., Tai Chi), physical activity (e.g., moderate-to-vigorous intensity aerobic exercise) and digital health and dietary health interventions have been shown to be effective at reducing depressive symptoms [17,18,19,20]. However, many of the interventions require cultural adaptations if they are to be practiced [21]. The implementation of certain non-pharmacological interventions (e.g., psychotherapy) might be compromised because these interventions demand highly trained specialists. Systematic reviews showed that exercise is effective to reduce depressive symptoms in many populations, including older people [22], adolescents [23], people with various chronic illnesses, such as major depressive disorder [24], and neurologic disorder [25]. In particular, aerobic exercises demonstrated a larger effect compared to other forms of exercises (e.g., stretching) [26]. The benefits of physical activity demands intensive physical activities, but adults with depression have barriers against exercises, most commonly exemplified by their lack of motivation, exhaustion, and lack of access to adequate exercising environments [27], plausibly explaining their high attrition rates [28]. However, psychoeducation can help people strengthen their coping strategies which are needed to manage stress and preserve mental health. A systematic review with a meta-analysis indicated that psychoeducation is both effective in reducing stress (d = 0.27) [29], and in helping people to manage stress due to trauma. There is a need to develop a cost-effective and culturally-adapted non-pharmacological intervention for African IDPs. A dance intervention refers to the use of body movements in response to musical sounds [30]. The intervention can be performed individually with a dance partner or in a group. Dance interventions have been proven to strengthen empowerment, acceptance and embodied self-trust [31], and to effectively reduce depressive symptoms [30]. Dance interventions have been found to effectively reduce depressive symptoms in adult populations [31]. A circle is frequently used in the profession of dance/movement therapy, and a circle dance may be using healing elements of dance/movement therapy [32]. An African Circle Dance (ACD) is an African traditional dance that is performed ceremonially as well as on other occasions, where a group of people dance in a circle with the accompaniment of drums [33,34]. ACD may be more readily acceptable to the general African population than other forms of using dance as a therapy [33,34].

Dance interventions have been shown in the literature to have favourable effects on many mental health symptoms (e.g., depressive symptoms) [30,35]. Dance intervention for the disadvantaged adults with depression was found to be acceptable (e.g., with opportunity to connect with others, subjects do not want it to stop) [36]. African dance is a conduit of individual and community healing as Africans conceptualize illness and health as an integration of the social, physical and mental realms: which all of these could be impacted by trauma [37]. Because of its potential healing effect, it is adopted and modified purposefully as a therapeutic modality to treat mental health symptoms in African people facing adversity. ACD is not novel as a ceremonial activity, but its effect on treating mental health symptoms in African IDP is novel. However, its healing effects on mental symptoms have never been examined among IDPs.

Folkman and Lazarus’ stress and coping theory is widely used and was chosen to guide this study because it is able to guide the development of many creative arts interventions for the management of stress and related emotional outcomes (e.g., anxiety and depression) [38,39]. Many research indicated that dancing intervention is effective in addressing moderate and mild depression but they are limited to high attrition rate in some studies [40]. IDPs often experience anxiety and stress from having to deal with unfamiliar environments, a loss of family, and previous trauma. Stress is known to be associated with negative affections (e.g., symptoms of anxiety and depression) [41]. Coping is an individual’s effort to manage internal/external demands by marshalling available resources [42]. When confronted by this stressor (e.g., displacement), IDPs choose available measures to decrease their stress and enhance their ability to cope. In this study, ACD was chosen as the intervention component because (a) it is widely accepted in IDP people, (b) could be applied to people with a traumatic experience, and (c) is in line with the needs for psychological rehabilitation in trauma-informed care as it evokes abnormal responses [40]. In this study, we employed psychoeducation and an ACD intervention as components to promote the psychological wellbeing of IDPs [43]. Psychoeducation is evidently an effective strategy to enhance psychological wellbeing [44]. For the content, we included the essential knowledge of daily stress management tips and available resources to seek necessary help to enhance the IDPs’ coping skills in the camps. Suggestions on positive health behaviour such as physical activities, environmental management, self-care, and other coping techniques were also made to enhance their psychological wellbeing (The contents of the psychoeducation will be further described below).

In the ACD intervention, dancing processes were employed to cope with the stressors. ACD comprises two major therapeutic components, which are body movement and music. They interplay with one another through dancing (e.g., tapping the rhythm of music) to stimulate interoception, which is the process of sensory information inside the body being transmitted and communicated to the brain and other body structures that occurs with or without conscious attention [45]. Interoception acts as distractors to shift a person’s attention away from the negative experience and replace it with positive thoughts [46]. This shifting of attention serves as one of the emotion-focused coping strategies allowing the person to cope with stress. As a result, stress and negative emotions could be healed [46]. There is evidence showing that dance interventions, including African dance, reduce physiological stress, perceived stress, and negative affect in various populations (e.g., older people and college students) [47,48].

## 2. Methods

A quasi-experimental design was adopted. Two IDP camps of similar size and run under the same government were chosen and randomised into an intervention group and a control group. This study was reported using the Transparent Reporting of Evaluations with Non-randomized Designs (TREND) reporting guideline [49] (Appendix A). The study was conducted between 7th December 2019 and 15th March 2020.

### 2.1. Ethical Considerations

Ethical approval was obtained from The Hong Kong Polytechnic University (HSEARS20190802005), the Borno State Ministry of Health (MOH/GEN/6679/1), and the State Emergency Management Agency (BO/SEMA/56/VOL.II/33). After being given an explanation of the study, eligible clients were invited to participate, giving their signed written consent. The participation of the clients was voluntary, and they were well-informed about the purpose, nature, and procedure of the study. The anonymity of the participants and confidentiality of their data were closely monitored.

### 2.2. Participants

Subjects were recruited using convenience sampling. All participants were recruited from their respective camp of residence. With the approval of the camp management office, the officers of the psychosocial support units of the IDP camps approached the potential participants and the officers refer them to the research assistants for eligibility screening. The principal researcher placed the information leaflet about the programme in places between the two centres in order to invite people to come for a dance programme screening. Potential participants were told about the ACD programme, potential risks and benefits, as well as their ethic rights (e.g., rights to withdraw) as stipulated on the consent form. The criteria for inclusion in the study were those: (1) aged ≥18 years, (2) who lived in one of the two IDP camps, (3) exhibited depressive symptoms, as defined by a Depression Anxiety Stress Scale (DASS-21) score of ≥20 [50], and (4) had good mobility, as defined by a score of 7 in the modified Functional Ambulatory Classification (mFAC) [51]. Exclusion criteria were those (1) with a history of disabling diseases such as chronic arthritis, primary pulmonary diseases (requiring long-term oxygen therapy), and cardiac diseases (e.g., end-stage heart failure), which might limit their ability to participate in dancing, and (2) who were unable to communicate in the Nigerian Hausa language.

The displaced individuals involved in this study lived with their family members in the Muna Garage and Teachers Village IDP camps [52,53]. They fled their homes because of an on-going ethno-religious crisis, and a significant proportion of them suffered from mental distress (e.g., depressive symptoms, stress [54]. The two camps each held between 31,019 and 39,560 people [53,55]. Both camps were managed by the State Emergency Management Agency (SEMA) and supported by Non-Governmental Organizations (NGOs).

### 2.3. Interventions

#### 2.3.1. Development of Psychoeducation Talks

Since psychoeducation has proven to be an effective strategy to help people cope with stress [29], we adopted it as the standard care for stress management. It was provided to both groups to ensure that subjects who were identified as having depressive symptoms would receive an evidence-based intervention to manage their psychological symptoms [56].

Psychoeducation was delivered by adopting the teaching contents of “10 stress Busters” developed by the National Health System because they were developed for laypeople to self-practice [57]. It does not require healthcare professionals, which are in shortage in IDP camps, to deliver. These items were grouped into (a) Positive health behaviour which includes: (1) avoid unhealthy habits, (2) work smarter, not harder, (3) challenge yourself, (4) be active, and (5) take control. (b) To provide support on how to cope and adjust their daily life in IDPs camp using the resources available: (1) connect with people, (2) have some ‘me time’, (3) help other people, (4) try to be positive, and (5) accept the things you can’t change [57]. They were also advised to seek professional help in terms of sickness or counselling using the camp clinic. To adapt these educational contents for people who might struggle with literacy, we incorporated pictures and posters to guide the teaching process. A mental health nurse delivered psychoeducation at the IDP camps to groups of 20–50 participants. We verbally met camp officers and told them our study plans (protocol plan for intervention, schedule of the programme and psychoeducational contents) were presented.

#### 2.3.2. Development of the ACD Dance Protocol

The ACD programme was developed through a two-stage process: (1) a systematic review of randomised controlled trials and (2) a Delphi process. In the systemic review, trials (*n* = 25) examining the effect of a dance intervention on depressive symptoms in various populations (e.g., dementia, breast cancer) were identified. Guided by the template for intervention description and replication (TIDieR) checklist (e.g., materials, dosage), components of the interventions were extracted from the identified articles [58]. Systematic reviews showed that an effective dance intervention is comprised of the following features: (1) administered by a dance therapist or specialist, (2) delivered face-to-face, (3) the dance load of each session ≥30 min, (4) 1–4 sessions per week, (5) and with a duration of 6–12 weeks [59,60]. We subsequently consulted a panel of experts and potential participants (*n* = 8), including professors with experience in designing a study with music or educational intervention (*n* = 2), a dance specialist (*n* = 1), a nurse (*n* = 1), a psychologist (*n* = 1), a psychiatrist (*n* = 1), and IDPs (*n* = 2). Finally, we revised the intervention protocol according to the panel’s comments and finalized the intervention protocol after achieving a full consensus in the second round of consultations through a Delphi process [61].

#### 2.3.3. Intervention Group

The participants of the intervention group received the same psychoeducation, in addition to a 75-min ACD session weekly for eight weeks. Table 1 showed the details of the ACD session: a 5-min briefing session, subjects were given the opportunity to introduce themsleves briefly, after which the day activity was introduced. At this stage, the physical condition of the subjects was seen, followed by a 10-min warm-up session of stretching exercises. A 50-min ACD session was then led by a dance specialist. During the dancing period, a band of musicians (i.e., two drummers and one flautist) employed Ganga Kura drums and an Algaita (i.e., a traditional African flute) to play the African music. The participants danced alongside the music in a circle made up of a group of participants. The dance type was called Maliki, which is a popular Kanuri (i.e., name of a tribe in North-Eastern Nigeria) traditional dance. The dance movements began at a point of a circle and it ended at the same point of the circle. The participants partnered with another participant of the opposite sex and danced together in the circle. The dancing movements involved multiple body parts (i.e., legs, hands, heads, fingers and torso). The rhythm is key to the dance movements and the participants dance in response to the music that it is like a conversation between the participants and music, and vice-versa. A 10 min cool-down session was followed. In this phase, the musical rhythm and dancing movement faded gradually towards the end. The same set dancing procedures (i.e., dancing pattern and music) repeated every week. A dance specialist is an expert with a definitive knowledge of a particular dance that is required by others, and was able to show essential dance standards by act demonstration [62]. The dance specialist, who is registered at the Borno State Ministry of Art and Culture, Nigeria, delivered the ACD intervention for the intervention group for the whole eight weeks. Eight sub-group sessions with 12–13 participants were conducted according to the protocol.

#### 2.3.4. Control Group

The participants in the control group received psychoeducation only. As shown in Table 1, each session lasted for 60 min, including a 5-min check-in process, a 40-min educational session, a 5-min clarification session, and a 10-min oral quiz session. The psychoeducation delivered by the mental health nurse was conducted on week 1 and week 4 (Table 2). The week 4 materials were the same as those of week 1 and were for the purpose of reinforcement.

#### 2.3.5. Trial Process

The whole ACD programme lasted for a period of 12 weeks, excluding two weeks baseline period. Eight weeks was used as the treatment phase, and four weeks follow-up period (Table 2). At week 1 and 4, psychoeducation was delivered. Outcomes were assessed at baseline, week 8 and 12. During the treatment phase, compliance, withdrawals, and adverse events/safety were assessed.

### 2.4. Objective

The objective of this study was, therefore, to evaluate the effects of ACD on the depressive symptoms, anxiety, and stress of IDPs who display depressive symptoms. We hypothesised that depressive symptoms, stress and anxiety reduce more in the intervention group than in the control group.

### 2.5. Outcomes

Demographic data (i.e., age, gender, education, employment, marital status, use of anti-depressive drugs) and clinical data (i.e., depressive symptoms, stress, anxiety, social support, and coping) were collected at T0. The primary outcome of this study was depressive symptoms, and the secondary outcomes were anxiety and stress. Trained research assistants collected data at baseline (T0), in the week immediately after the completion of the 8-week intervention (T1), and four weeks after the completion of the intervention (T2).

A Hausa-language version of the Depression Anxiety Stress Scale—short form (DASS-21) was employed to measure the three outcome variables because it comprises of three constructs (i.e., depression, anxiety, and stress) and 21 items [63]. Each construct is measured by seven items and each item assessing an emotional state is quantified using a 4-point scale (i.e., 0 = never, and 3 = almost always). Each subscale score of DASS-21 ranged from 0 to 21. To obtain a score similar to the DASS—full form which is comprised of 42 items, each DASS-21 subscale score was doubled so that each subscale score ranged from 0 to 42 [64]. The depression subscale score was interpreted as normal (0–9), mild (10–13), moderate (14–20), severe (21–27), and extremely severe (28+); the anxiety subscale score was interpreted as normal (0–7), mild (8–9), moderate (10–14), severe (15–19), and extremely severe (20+); the stress subscale score was interpreted as normal (0–14), mild (15–18), moderate (19-25), severe (26–33), and extremely severe (34+) [65]. DASS-21 was found to have good internal consistency (α = 0.82–0.93) and inter-rater reliability (ICC = 0.82–0.86) [66]. With regards to convergent and discriminant validity, the DASS-21 has a positive correlation with Beck’s Anxiety Inventory (BAI) and Beck’s Depression Inventory (BDI) (r = 0.5–0.8) [67]. It is negatively associated with positive affectivity and positively correlated with negative affectivity scales [68]. The DASS-21 scale has an inverse correlation with quality of life measures [69]. The concurrent validity of the DASS-21 is good as it shows adequate discrimination of both clinical and non-clinical samples and other specific groups [70]. This tool was translated into the Hausa African language, culturally adapted, and psychometrically tested. It was found to have an overall internal consistency of 0.8, and 0.7 for stress, depression, and anxiety, respectively [63].

### 2.6. Sample Size

The sample size was estimated using GLIMMPSE, based on an a priori power analysis employing a general linear mixed model [71]. We estimated the effect size from a similar study that evaluated the effect of a dance intervention on depression in the intervention group at T0 (mean = 16.00, SD = 12.35) and T1 (mean = 8.76, SD = 9.64); and in the control group at T0 (mean = 18.62, SD = 11.98) and T1 (mean = 16.92, SD = 9.74) [72]. Given that the attrition rate of a similar study was about 10% at 3 months [72], the minimum total sample size needed was 182 at a 0.05 level of significance and 0.8 power.

### 2.7. Assignment Methods

This study was conducted in two IDP camps: Muna Garage and Teacher’s Village, North-Eastern Nigeria. One camp was assigned to one single group to prevent effect contamination within the same venue. The group assignment was determined by simple randomisation using the coin flipping method [73]. Muna Garage camp was assigned as the intervention group and Teacher’s Village camp as the control group.

### 2.8. Blinding

The outcome assessors were only blinded to the group label of the study because it was not possible to blind the interventionist and participants. Due to the nature of the intervention, it was difficult to blind the participants and the interventionists.

### 2.9. Unit of Analysis

Groups of individuals were assigned to study conditions. (i.e., intervention and control groups). The analyses were performed at the group level, where mixed effects models (via generalised estimating equations) were employed to account for random subject effects within each group.

### 2.10. Statistical Methods

We used IBM SPSS statistical software version 20 (IBM Corp, Armonk, NY, USA) to analyse the data. The demographic and clinical profiles of the participants at baseline were described using mean with standard deviation and frequency with percentage according to their level of measurement. The balance of the baseline characteristics was compared through either a Chi-square or independent t-test according to their level of measurement. Outcomes at different time points were described by groups using mean and standard error.

To evaluate the effects of the ACD programme, generalized estimating equations (GEE) were employed. The dependent variables were depressive symptoms, stress, and anxiety. The independent variables were groups (i.e., intervention vs control), time (i.e., baseline, week 8, and week 12), and group x time. All equations were adjusted for potential confounding factors, which were determined by two principles: (1)Demographic variables that are known to be associated with dependent variables in the literature (i.e., age, gender, education, marital status, and employment) [74], and(2)Outcome variables (i.e., depression, anxiety, and stress) that were not balanced between groups at baseline because they are known to be associated [75].

The results of both adjusted values were reported. Due to unbalanced characteristics between groups, we adjusted the equations because this was not a randomised controlled trial in which potential confounding factors could alter the effects of the intervention on the outcomes. We set the level of significance at 0.05. The effect size was reported using the Cramer’s V [76]. The score was interpreted as very weak (0–0.04), weak (0.05–0.09), moderate (0.10–0.14), strong (0.15–0.25), and very strong (>0.25) [77]. Missing data were handled using an inverse-probability weight GEE if the assumption of missing-at-random was fulfilled [78].

## 3. Results

### 3.1. Participant Flow

As can be seen in Figure 1, 280 subjects were approached for screening. Fifteen of them were not eligible, and 67 declined to participate. In the end, 198 participants were recruited and allocated to either the experimental (*n* = 100) or the control (*n* = 98) group. The recruitment rate was 70.7% (i.e., 198/280). All of the subjects received the interventions and data collection was completed at T1. The attrition rate was 17% (*n* = 17) for the intervention group and 10.2% (*n* = 10) for the control group at T2. All the eight dance subgroups have completed their sessions 100%. However, subjects did not attend to all the sessions. Therefore, the average dance attendance per subject in the intervention group was 79.5%; and the overall attendance rate for the intervention group is 74.9%. For the control group with only psychoeducational intervention, all subjects attended the two sessions at week 1 and 4 (i.e., 100%). We only have missing data at T2 in twenty-seven subjects on the outcome variables. The reasons for the loss of follow-up were “change of camp” (*n* = 25) and “being arrested” (*n* = 2). The analysis included all subjects as assigned to the intervention and control groups, respectively.

### 3.2. Recruitment

The recruitment period was during 21st November 2019 to 5th December 2019 and the follow-up period was during 9th to 15th March 2020.

### 3.3. Baseline Data

#### Demographic and Clinical Profiles at Baseline

As shown Table 3, the majority of participants were 18–29 years of age (*n* = 102, 51.5%), female (*n* = 127, 64.1%), unemployed (*n* = 184, 92.9%), uneducated (*n* = 147, 74.2%), and married (*n* = 114, 57.6%). Less than 10% of the subjects were taking an anti-depressant (*n* = 19, 9.6%).

### 3.4. Baseline Equivalence

The baseline group differences were assessed using the Chi-square test and independent t-test for categorical and continuous variables. The groups were not balanced in terms of education (*p* = 0.038), marital status (*p* < 0.001), and depression (*p* < 0.001). However, there was no group differences for gender (*p* = 0.582), age (*p* = 0.195), employment status (*p* = 0.174), and use of anti-depressant (*p* = 0.774) (Table 3).

### 3.5. Numbers Analysed

The analysis of ACD effects included all subjects that completed the interventions at T1. Although we lost 27 subjects at follow-up, they were all included in the analysis (i.e., 100 for the intervention group and 98 for control).

### 3.6. Outcomes and Estimation

#### Effects of the ACD Intervention

As shown in Table 4, in the adjusted model, the depressive symptoms reduced after the completion of the intervention at both T1 and T2 significantly in both groups. The reduction of depression in the intervention was larger than that of the control group with a very strong group*time interaction effect (*v* = 0.33, *p* < 0.001). The reduction was greater in the intervention group than in the control group at both T1 (β = −6.6, 95% CI = −8.73, −4.41) and T2 (β = −3.4, 95% CI = −5.46, −1.39). The interpretation of results did not differ between the adjusted and unadjusted models.

As shown in Table 4, in the adjusted model, the anxiety reduced after the completion of the intervention at both T1 and T2 significantly in both groups. However, the reduction of anxiety in the intervention was not significantly different from that of the control group with a very weak group*time interaction effect (*v* = 0.08, *p* = 0.315). The interpretation of the results did not differ between the adjusted and unadjusted models.

As show in Table 4, in the adjusted model, the stress reduced after the completion of the intervention at both T1 and T2 significantly in both groups. The reduction of stress in the intervention was larger than that of the control group with a strong group*time interaction effect (*v* = 0.15, *p* = 0.008). The reduction was greater in the intervention group than in the control group at T2 (β = −3.3, 95% CI = −5.45, −1.14), but not at T1. The interpretation of results did not differ between the adjusted and unadjusted models.

### 3.7. Adverse Events

No adverse effects (e.g., injuries and falls) were reported.

## 4. Discussion

### 4.1. Interpretation

To the best of our knowledge, this is the first study to investigate on the effects of an ACD programme (i.e., ACD in addition to psychoeducation) on mental health outcomes (i.e., depressive symptoms, stress, and anxiety) among African IDPs. It is also the first study to enquire whether ACD would enhance the effects of psychoeducation. This study showed that the intervention could reduce depressive symptoms from severe to mild level, reduce anxiety from extremely severe to moderate level, and reduce stress from moderate level to normal. The effects in the intervention group were very strong in depressive symptoms, and strong in stress than those in the control group. However, the effect in anxiety was very weak.

ACD is a form of dance intervention. The literature suggests that dancing, including African dance, is effective in reducing stress, as measured by perceived stress and salivary cortisol [79]. Previous studies revealed that a reduction in stress is strongly associated with a reduction in depressive symptoms [75]. The findings of this study provide preliminary support for the view that ACD possibly reduces stress and depressive symptoms through a mechanism different from that employed in psychoeducation, since this study showed that ACD yields effects beyond psychoeducation. In addition, the music component has the ability to adjust emotional states and promote mind–body interactions and is widely accepted for promoting relaxation in healthcare. As an exercise with moderate intensity, dance movements may release endorphins that help people feel relaxed after exercise [80]. The mechanism for stress and emotional adjustment could be that the dance and music elements might regulate states of emotions through mind-body interactions which occur due to the effects of the released neurotransmitters making participants feel entertained and happy [81,82,83]. Interoception is another critical maturational element of dance whose effects might play a role in mood regulation and the promotion of the emotional state [46]. ACDs’ elements of music and body movement stimulate interoception and act as distractors to shift a person’s attention away from negative experiences and replace them with positive thoughts [40,41,84]. The group element of the dance on its own also has a positive curative effect on the participants while moving in a circle; the therapeutic factors of the group might advance exchange, universality, instillation of hope on an individual, activates collective unconsciousness, corrective recapitulation and mirror reaction [32]. The group effect was found to have a short-term effect on the global values which fosters social integration [32,85]. Another possible explanation for stress reduction in the current study is that the ACD intervention was undertaken in the social environment [85]. Further studies should purposefully examine the mediation effect of stress between dancing and depressive symptoms. Clinically, an ACD programme (i.e., ACD plus psychoeducation) holds promise to contribute in the management of stress, and depressive symptoms in IDPs.

This study demonstrated that psychoeducation is effective in reducing anxiety, and that dance does not exert any additional effects. Unlike anxiety disorder, post-traumatic stress is associated with a different circuitry in the brain with diminished responsivity in the anterior cingulate cortex [86]. People who have undergone a traumatic experience might not benefit in terms of anxiety from the therapeutic mechanism employed by a dance intervention. Further studies should clarify the effects of dance interventions on anxiety in people with traumatic experiences.

The ACD was highly feasible with high recruitment and low drop-out rates. The acceptability of the ACD in terms of attendance rate was also high. However, the level of satisfaction and barriers of the ACD were not measured in this study. Possible barriers may include space and security. In our study, the camp sites were highly spacious and there were limited number of terrorist attacks during the intervention period. However, some other camp sites may not provide adequate space for group dancing. Terrorist attacks could be more often that the continuity of the programme might be hindered. Further studies should explore the level of satisfaction and barriers of ACD to be implemented in the IDPs.

### 4.2. Generalizability

There are several limitations to this study. First, it was not a truly randomised controlled trial because of the non-random subject allocation. It is a quasi-experimental design, not a true randomised controlled trial due to the consideration of subjects’ contamination in the IDP camps. Due to the non-random subjects’ allocation to the groups, it was challenging to know how well the IDPs population were represented, limiting generalisability of this study to other populations with traumatic experiences outside Nigeria [87]. Second, the lack of balance in the baseline characteristics of the subjects could indicate that unobserved confounders might have affected the level of confidence in the effect of the dance intervention, although the models were adjusted for known unbalanced co-variates observed at baseline. The baseline scores showed higher depressive symptoms in the intervention group compared to the control group. Sub-analysis proved that there was a regression to the mean, in particular with those in the intervention group who showed a greater reduction because they were higher at baseline. There is a risk that the treatment effect is overestimated by solely looking at changes in scores [88]. Third, the use of a specialist, not a dance/movement therapist was adopted due to resource constraint in Africa. We, therefore recommended the employment of a registered specialist with experience in leading African dance for future studies. The effect could possibly be due to the amount of attention between groups because there was no interaction in the control group after psychoeducation. Without a control group of usual care, the anxiety-reducing effects observed in the psychoeducation group might possibly be caused by other confounding factors. The intervention of this study was not delivered by dance therapists. Nonetheless, this protocol delivered by the dance specialist also demonstrated promising therapeutic effects. We adopted the “10 Stress-buster” as the psychoeducation contents. Although the contents were developed by National Health System, its effects have not been empirically evaluated. This study, therefore, could not confidently conclude that ACD is more superior to psychoeducation. Therefore, in future studies, we recommend that a cluster randomised controlled trial should be employed to ensure for the comparability between groups; one more passive control group should be added to provide better understanding of the effects between ACD and psychoeducation, as well as that efficiency and cost effectiveness of the ACD delivered by dance specialists and dance therapists should be compared.

### 4.3. Overall Evidence

Compared to psychoeducation, ACD has additional effects on depressive symptoms and stress but not on anxiety in African IDPs with traumatic experience. ACD might be a valuable complementary intervention in health promotion because it may be more readily acceptable to general African population. An ACD programme (i.e., ACD in addition to psychoeducation) holds promise to contribute to the management of stress for people living in IDPs with traumatic experiencess. Further studies should clarify how this ACD programme could be generalised to other IDP camps or other healthy people in Africa.

## Figures and Tables

**Figure 1 ijerph-18-00843-f001:**
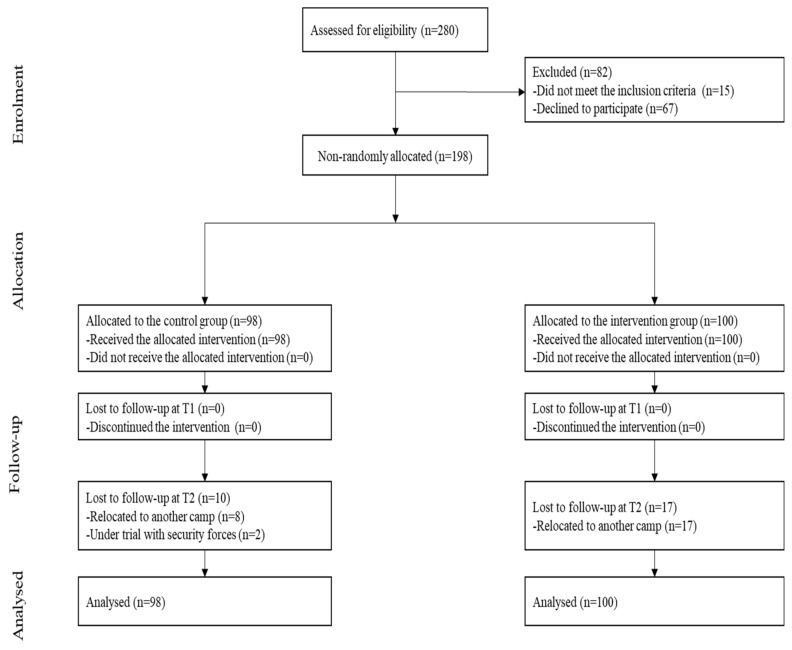
Flowchart.

**Table 1 ijerph-18-00843-t001:** ACD Programme.

Time	Session Name	Contents
Psychoeducation (60-min)		
5 min	Check-in session	Self-introduction, activity introduction
40 min	Education session	Lecture on facts relating to the psychological effect of stress, how to cope in an IDP camp, resources available in the IDP camp, and coping skills depicted on the “10 stress Busters” [57].
5 min	Clarification session	Answer questions raised by the participants
10 min	Oral quiz session	Evaluate the participants’ knowledge
Dance intervention (75-min)		
5 min	Check-in session	Self-introduction, activity introduction
10 min	Warm-up session	Preparation for the dance
50 min	ACD session	Dance exercise under the teaching and supervision of a dance specialist
10 min	Cool-down session	Preparation to end the dance

ACD: African Circle Dance.

**Table 2 ijerph-18-00843-t002:** Trial Process of the ACD programme.

Tasks	Baseline	Treatment Phase								Follow-Up
		Week								
	−2 to 0	1	2	3	4	5	6	7	8	9 to 12
Informed consent	X									
Obtain participant’s Assent	X									
Intervention										
ACD Dance		X	X	X	X	X	X	X	X	
Psychoeducation		X			X					
Control										
Psychoeducation		X			X					
Outcomes										
#Depression	X								X	X
#Anxiety	X								X	X
#Stress	X								X	X
Process monitoring										
Compliance		X	X	X	X	X	X	X	X	
Reasons for withdrawal		X	X	X	X	X	X	X	X	
Adverse events/safety		X	X	X	X	X	X	X	X	

ACD: African Circle Dance, #measured by Depression, Anxiety Stress Scale (DASS-21).

**Table 3 ijerph-18-00843-t003:** Demographic and clinical characteristics.

Participants’ Characteristics	All (*n* = 198)	Control (*n* = 98)	Intervention (*n* = 100)	*p*-Value
Age, n (%)				
18–29 years	102 (51.5)	60 (61.2)	42 (42.0)	0.195
30–49 years	80 (40.4)	32 (32.7)	48 (48.0)	
50 years above	16 (8.1)	6 (6.1)	10 (10.0)	
Gender, n (%)				0.582
Male	71 (35.9)	37 (37.8)	34 (34.0)	
Female	127 (64.1)	61 (62.2)	66 (66.0)	
Education, n (%)				
No formal education	147 (74.2)	65 (66.3)	82 (82.0)	0.038 *
Primary education	19.0 (9.6)	13 (13.3)	6 (6.0)	
High school or above	32.0 (16.2)	20 (20.4)	12 (12.0)	
Marital status, n (%)				<0.001 **
Married	114 (57.6)	49 (50.0)	65 (65.0)	
Widowed	25 (12.6)	9 (9.2)	16 (16.0)	
Divorced	22 (11.1)	10 (10.2)	12 (12)	
Never married	37 (18.7)	30 (30.6)	7 (7.0)	
Employment, n (%)				0.174
Employed	8 (4.0)	5 (5.1)	3 (3.0)	
Unemployed	184 (93.0)	88 (89.8)	96 (96.0)	
Retired	6 (3.0)	5 (5.1)	1 (1.0)	
Anti-depressant, n (%)				0.774
Yes	19 (9.6)	10 (10.2)	9 (9.0)	
No	179 (90.4)	88 (89.8)	91 (91.0)	
Retired	6 (3.0)	5 (5.1)	1 (1.0)	
DASS-21, mean (SD)				
Depression	26.5 (4.1)	25.2 (3.6)	27.7 (4.2)	<0.001 **
Anxiety	23.4 (6.9)	23.2 (8.1)	23.6 (5.6)	0.645
Stress	21.5 (6.5)	21.8 (6.8)	21.2 (6.1)	0.532

* *p* < 0.05, ** *p* < 0.001; DASS-21 scale: Depression Anxiety Stress Scale-21 SD: Standard Deviation.

**Table 4 ijerph-18-00843-t004:** Effects of the ACD intervention on depression, anxiety, and stress.

Variables	Time	Control (*n* = 98) Mean (SE)	Intervention (*n* = 100) Mean (SE)	GxT Interaction Effects β (95% CI)	*p*-Value
Depressive symptoms—Unadjusted					
	T0	25.2 (0.4)	27.7 (0.4)		
	T1	13.8 (0.7) *	9.7 (0.7) *	−6.6 (−8.73, −4.41)	<0.001
	T2	12.7 (0.8) *	11.6 (0.4) *	−3.5 (−5.51, −1.47)	0.001
Depressive symptoms—Adjusted					
	T0	24.8 (2.7)	26.1 (2.6)		
	T1	13.4 (2.7) *	8.1 (2.7) *	−6.6 (−8.73, −4.41)	<0.001
	T2	12.3 (2.7) *	10.2 (2.8) *	−3.4 (−5.46, −1.39)	0.001
Anxiety—Unadjusted					
	T0	23.2 (0.8)	23.6 (0.6)		
	T1	13.9 (0.8) *	15.8 (0.7) *	1.4 (−1.53, 4.26)	0.355
	T2	12.4 (0.8) *	14.9 (0.6) *	2.0 (−0.74, 4.71)	0.153
Anxiety—Adjusted					
	T0	21.3 (4.4)	21.5 (4.3)		
	T1	12.1 (4.5) *	13.6 (4.3) *	1.4 (−1.53, 4.26)	0.355
	T2	10.5 (4.5) *	12.7 (4.4) *	2.0 (−0.69, 4.75)	0.143
Stress—Unadjusted					
	T0	21.8 (0.7)	21.2 (0.6)		
	T1	5.5 (0.3) *	3.0 (0.3) *	−1.9 (−3.94, 0.11)	0.064
	T2	9.6 (0.5) *	5.7 (0.4) *	−3.3 (−5.46, −1.15)	0.003
Stress—Adjusted					
	T0	21.4 (3.6)	20.5 (3.7)		
	T1	5.2 (3.6) *	2.6 (3.5) *	−1.9 (−3.94, 0.11)	0.064
	T2	9.3 (3.6) *	5.3 (3.6) *	−3.3 (−5.45, −1.14)	0.003

* Significant; M: Mean; SE: Standard Error; β (Beta coeficient); GxT: Group*Time; CI: Confidence Interval.

## Data Availability

The data presented in this study are available on request from the corresponding author. The data are not publicly available due to ethical restrictions.

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
