# Peer review of "Effects of an African Circle Dance Programme on Internally Displaced Persons with Depressive Symptoms: A Quasi-Experimental Study"

_ijerph, 2021, doi:10.3390/ijerph18020843_

Round 1
Reviewer 1 Report
The authors report on a novel and culturally appropriate intervention for addressing stress, anxiety, and depression. The intervention is highly innovative. Enthusiasm for the study is tempered by the theoretical bases, the study design, and reported data that do not seem to match in the tables.
The stress and coping theory does not adequately explain why the investigators proposed the intervention. How does dancing reduce perceived threat or increase coping resources? Moreover, there is a wealth of research on physical exercise as highly effective in addressing moderate and mild depression. One of the difficulties is that there tends to be high attrition in some studies of exercise interventions for depression.
A study strength is the standard of care comparison group. However, there was a major difference in dose of the intervention between the groups; hence, the differences in the outcomes may be due, in part, to the amount of attention. The authors should indicate the average number of sessions attended by each group. Did the authors assess any non-western medicine that the participants may have been taking for their mental health? Given the weakness of not using a randomized clinical trial and the lack of equivalence between groups at baseline, it may be useful to examine those who were high versus low in depression at baseline in level of depression as one explanation may be that there was regression to the mean with those in the experimental group showing a greater reduction because they were higher at baseline.
Who approached potential participants? What were they told about the program when they were recruited?
The authors should provide more information about both barriers to implementing the intervention and reactions to the intervention, e.g., did participants find it to be culturally appropriate? Is this intervention something that can be easily disseminated and sustained in the IDP camps? Is African Circle Dance a salient component of the participants’ culture, or was this novel to them?
The authors should temper their language on causality. Rather than saying, “This study showed that an ACD programme significantly reduced stress and depressive symptoms,” state that those in the experimental group reported a greater reduction in depressive symptoms than those in the standard of care. The statement that the psychoeducational group reduced anxiety is also problematic as it may have been reduced due to other factors.
The greatest concern about the study is that the data presented in tables 2 and in table 3 do not match. There may be a good explanation for these differences; perhaps table 3 is adjusted, but it is not apparent to the reader and hence calls into question the validity of all of the data.
The authors should recommend in more detail future research based on their findings and observation. Is a specialist needed to implement this ACD intervention?
Was the duration after the end of the intervention for T1 the same for both conditions?
Reviewer 2 Report
Thank you for initiating this study, this is an important topic that deserves attention, and not least for this target group. The creative and social aspect of dance together with the effects linked to the physical activity and music can offer valuable mental (and physical) benefits for this people in this situation.
Overall, I suggest the authors to consider the following:
- Literature review should be updated and also expanded
- Parts of the conclusion needs to be rephrased
- To improve the paper, please see the attached file and/or below. I have stated my recommendations as comments in the attached PDF. Every yellow-marked section has an attached comment. If it for any reason is troublesome to read them in the PDF, I have also added this input below, marked with the row-numbers. Please forgive me for many comments, I hope they will be of use to you:
--------
4 Suggestions regarding the title: -Effects instead of Effect -Add design
12 The text in the background has to be in line with the text in the abstract.
25-26 I would recommend to replace this sentence with a condensed summary from the actual conclusion in the manuscript instead.
I suggest a more humble approach, ACD could be a valuable complementary intervention in health promotion but more research is needed.
41-42 I would recommend a deeper dive in recent literature and update reference no 6 and 7.
42 Please check the numbers - the abstract has to be in line with the text in the manuscript. (Abstract: Depressive symptoms in IDPs in Africa are prevalent, at over 30%.)
47 True, but I recommend to delete the word 'therefore'. They are preferable for other reasons aswell (easy-access, health economic benefits, positive life habits and so on)
48 I recommend adding cognitive behavioral therapy (CBT) to this list as an example of psychotherapy.
54-55 Suggest to rephrase to: However, psychoeducation can help people to strengthen coping strategies needed to manage stress and preserve mental health.
60 Ref no 17 does not give a definition of dance intervention, but is a valuable reference for understanding the effects that a dance intervention can give. Suggest something like:
Dance interventions have been found to strengthen empowerment, acceptance and embodied self-trust (17) and to effectively reduce depressive... (18)
68 Also look in to these, maybe you want to add some of them to your list (upgrade from 2011 and reflect on the choice of words; dance movement therapy or dance intervention)=
- Schwender TM, Spengler S, Oedl C, Mess F. Effects of Dance Interventions on Aspects of the Participants' Self: A Systematic Review. Front Psychol. 2018; 9:1130.
- Meekums B, Karkou V, Nelson EA. Dance movement therapy for depression. Cochrane Database Syst Rev. 2015; 2:CD009895. doi: 10.1002/14651858.CD009895.pub2.
69 Add your hypotheses
95 Would prefer to rephrase to something like: Lazarus and Folkman’s stress and coping theory is widely used [29] and was chosen to guide this study because it applies to... / is in line with... (with ref)
103 I suggest that you add some references that combine these two modalities or similar
104 needs reference
104-106 This section raises many questions to the reader.
('Essential knowledge'?) Is it two separate interventions ("also")? Please add "further described below" - if it is the current psychoeducation you are referring to here
108 (Perhaps reconsider the word 'artistic'? )
112 Would recommend to add this reference here:
Duberg A, Jutengren G, Hagberg L, Moller M. The effects of a dance intervention on somatic symptoms and emotional distress in adolescent girls: A randomized controlled trial. Journal of International Medical Research 2020;48(2) 1–12. DOI: 10.1177/0300060520902610
116 check language
118 / Table 1 = This needs adjustment to get it in line with the text in Table 1 where it says:
"Lecture on facts relating to the psychological effect of stress, how to cope in an IDP camp, resources available in the IDP camp, positive health behaviour " Please describe how you communicated the adjustments to the IDP-camp.
119 This reference needs editing in the reference list.
(Maybe refer to it as a web-page?
https://www.nhs.uk/conditions/stress-anxiety-depression/reduce-stress/ )
Also; why was this chosen? What is the evidence for these 10 busters? Please add to the text if it has used before in similar studies to a similar target group.
125 Add "see Table 1."
145 Confusing with "The health talk" and the "Psychoeducation Talks" - please make it more clear - is psychoeducation a part of health talk or how do you mean?
148 Table 1: This has to be in line with the text above in 2.3.1. Development of Psychoeducation Talks . Please describe how you communicated the adjustments to the IDP-camp.
152 please add "for 8 weeks" to be more clear.
154 An explanation of how the dance was performed needs to be added to the text here. Describe the dance and if the dance style/theme varied during the 8 weeks, intensity, what kind of music etc.
155 eight weeks twice weekly is 16 times? needs clarification?
193 move to the section below
234 please describe
237 Add a figure text under ‘ Figure 1. Flowchart ’
285 ACDs´
285 At least add 'stimulates interoception' and act as ...
Maybe develop the sentence to two sentences regaring interoception?
Perhaps you will find it valuable to look in to these references=
- Sevinc G, Holzel BK, Hashmi J, Greenberg J, McCallister A, Treadway M, et al. Common and Dissociable Neural Activity Following Mindfulness-Based Stress Reduction and Relaxation Response Programs. Psychosom. Med. 2018.
- Schmalzl L, Crane-Godreau MA, Payne P. Movement-based embodied contemplative practices: definitions and paradigms. Front Hum Neurosci 2014;8:205.
288 Replace the reference 55 (to old) to at least two recently published and updated references - exercise and stressreduction is a wellstudied area and it is beneficial for your credibility as authors in the field if you update this section.
290 I suggest that you replace this sentence with a discussion on how the social aspect of the dance intervention adds to the results of this study. The fact that the dance was performed in a group is central and deserves to be included when discussing mechanisms.
Something like "A possible explanation for the observed stressreduction in the current study is that the ACD intervention was undertaken in a social environment”. ...followed by references from the current target group
293 Suggest change to something like: "holds promise to contribute in the management of stress and...
300 + The baseline scores showed higher depressive symptoms in the intervention group compared with the control group, and therefore can improve more. There is a risk that the treatment effect is overestimated by looking at change scores. (Vickers AJ, Altman DG. Statistics notes: analysing controlled trials with baseline and follow up measurements. BMJ. 2001;323(7321):1123-1124.)
+ And then discuss regression?
300 Please rephrase.
"due to the non-random subject allocation, there is a risk that ... " (and then mention1-3 risks).
311 see earlier comments

Reviewer 3 Report
The article has significant strenghts:
- The research question is novel and original for this target population
- Writing is perfectly clear, providing details about methods and measurements
- Well described Participant flow, good choice of statistical method
Limitations of the study
- could be seen in its quasi-experimental design.
However, the design is clearly described. It is a pragmatic trial, which can realistically be replicated in following studies.
- Generalizability of the study to other populations with traumatic experiences outside Nigeria may be limited.
However, the proces of development of the Dance intervention is well described, and may be adapted for IDP groups outside Nigeria, respecting local cultural specifics during implementation process.
This text is describing strictly quantitative outcomes.
Suggestion: I would like to ask the authors, whether they have collected also qualitative data reflecting the experience of participants. E.g. field notes, quotes, focus groups, etc. It would be interesting to mention such reflection in the the discussion section, or even in a following article.
In conclusion, the text is well written and I surely recommed acceptation of the article.
Reviewer 4 Report
Thank you for this paper. I appreciate the study and have only a few comments here:
Introduction:
- I have problems with this sentence: "A circle dance involves people dancing as a group in a circle,which symbolizes totality, wholeness, and completeness, and is a form of dance movement therapy" (Lines 61--63). A circle dance is not a form of dance/movement therapy. Instead, a circle is frequently used in the profession of dance/movement therapy and a circle dance may be using healing elements of dance/movement therapy.
- I would also caution the sentiment in this sentence:" Because of its cultural relevance, specificity, and suitability, ACD may be more readily acceptable to the general African population than other forms of dance therapy." (Lines 65-66) - Instead I would say "...than other forms of using dance as therapy"
- I would really like to learn more about ACD. Please go into greater detail about it. As this is the main intervention it is important for the reader to fully comprehend what this looks like.
Research Design:
- What I do not like about the research design (and this is a limitation which should be noted in the discussion section) is that the control group does not receive the intervention at all. It would be ethical to offer the intervention after the study to those who did not get a chance to experience it.
Methods:
- What studies were used for the systematic review (please change spelling, currently it says systemic review, line 127)?
- It does not appear that a dance/movement therapist was included in the consultation of experts (line 135-137). This is curious, given the list of essential features of a dance intervention in the previous sentence (line 132-133) that starts with 1)administered by a dance therapist. This is a shortcoming which should be acknowledged.
- Can you please clarify what a dance specialist is? (Line 154)
- I am confused about the length of the study. Throughout the text (and in the abstract) it says it was an 8-week intervention. However, in the methods section (Line 155-157) you say it was 8 group sessions which were conducted twice per week - this would make the study only be 4 weeks long. Please clarify.
- Again, greater detail about the ACD would be necessary here. You give more details about the psychoeducation group consisting of the "10 Stress Busters", but there is very little clarity about how the ACD looked - what music was used (I presume drumming)? Who did the drumming, how was the warm up different from the dance "exercise" (see table), etc.
- Did the same dance specialist conduct all 8 x 8 (or 16 - this is not clear) sessions?
- I would move the ethical considerations, including consent to the beginning of the methods section.
Results:
- This section is fine, although I do wonder about a specific pre-post measure of those 10% who took antidepressants.
Discussion:
- The reference "54" does not seem appropriate here since it is about "easy listening" music, not drumming. Again, more details about the employed music as well as the movements would be really helpful to determine how these factors positively affected depression.
- I wonder about the importance of group vs. individual movement. The idea of belonging and being part of an expressive moment. I would like for the authors to write about this aspect as well, as it is distinct from individualized self expression.
- The topic of psychological trauma is not given enough room in this paper. Please elaborate both in introduction and discussion as this definitely plays a big part in how the sessions were conducted and received (trauma-informed)
Round 2
Reviewer 1 Report
The authors were responsive to the majority of the prior comments. However, two remain.
Although the authors point out the problems with treating depression with exercise, they do not adequately represent its value as reported in the literature. Meta-analyses suggest that exercise is as effective as many other forms of treatment for depression.
Although barriers were not assessed, the authors could speculate about barriers to implementing the intervention. Did the authors assess any measure of satisfaction or acceptance of the intervention? If not, then in the limitations section, they should acknowledge that they have no information or feasibility or acceptability of this type of intervention.
Author Response
Please refer to our responses to your comments and suggestions in the attached pdf file.

Reviewer 2 Report
Thank you for a generously performed update of the manuscript. This in an interesting topic.
My concerns:
The reference list contains errors and gaps.
Overall, I would recommend further editing of English language.
In the introduction, reference no 5 is over-used.
Row 63-71 contains repetition.
Row 77 reference no 18 - I think you mean 23? And 17 - I think you mean 22?
I would recommend using a reference-system.
Author Response
Please refer to the attached pdf file for our responses to your comments and suggestions.
